# Infrared Single-Frame Small Target Detection Based on Block-Matching

**DOI:** 10.3390/s22218300

**Published:** 2022-10-29

**Authors:** Yi Man, Qingyun Yang, Tao Chen

**Affiliations:** 1Changchun Institute of Optics, Fine Mechanics and Physics, Chinese Academy of Sciences, Changchun 130033, China; 2University of Chinese Academy of Sciences, Beijing 100049, China; 3Academy for Advanced Interdisciplinary Studies, Northeast Normal University, Changchun 130024, China; 4Laboratory of Pinghu (Beijing Institute of Infinite Electric Measurement), Pinghu 314200, China

**Keywords:** infrared image, block-matching, small target detection, low-rank matrix recovery

## Abstract

The robust detection of small targets is one of the crucial techniques in an infrared system. It is still a challenge to detect small targets under complex backgrounds. Aiming at the problem where infrared small target detection is easily disturbed by complex backgrounds, an infrared single frame detection method based on a block-matching approach is proposed in this paper. Firstly, the input infrared image is processed by extracting blocks from it. A new infrared model is constructed by finding blocks that are similar to each such block. Then, the small target detection based on the block-matching model is formulated as an optimization problem of recovering low-rank and sparse matrices, which are effectively solved using robust principal component analysis. Finally, the results of processing are reconstructed to obtain the target and background images. A simple segmentation method is used to segment the target image. The experimental results from the actual infrared sequences show that the proposed method has better background suppression ability under complex backgrounds and better detection performance than conventional baseline methods.

## 1. Introduction

Infrared detection technology has many characteristics, such as anti-electromagnetic interference, a wide temperature range, and real-time observation. It has since been widely used in various applications, especially in the military and remote sensing [1]. Generally, small targets have a tiny proportion of pixels in infrared images. Meanwhile, target message loss is profound because of the low signal-to-noise ratio and poor contrast [2]. Background clutter and cloud also will increase the difficulty of infrared small target detection [3]. Therefore, infrared small target detection technology has great significance and is still a challenging study. The goals of researchers have been better real-time performance, a higher probability of detection, and a lower false-alarm rate.

Many scholars have recently proposed many effective methods to detect small targets. Currently, the practices of infrared small target detection can be divided into detection before track (DBT) and track before detection (TBD). Typical TBD methods include the dynamic programming method [4], the particle filter method [5], 3D matched filtering [6], and the pipeline filter method [7]. However, TBD methods combine the time and space information of sequence images to predict the target trajectory, which is challenging to apply in practice. In contrast, DBT methods detect small targets on a single frame, which have the advantages of fast detection speed and a high robustness to noise. The DBT methods can be summarized into three categories: the methods based on target features, the methods based on background features, and the methods based on low-rank sparse matrix recovery.

For methods based on target features, local information differences can be recognized in the human visual system [8]. Therefore, many small target detection methods based on the visual system are proposed [9,10,11,12,13,14,15,16]. Local Contrast Measure (LCM) is a usual method [9]. It combines the contrast mechanism of the human visual system with the derived kernel model to achieve target enhancement and background clutter suppression. However, LCM makes an excessive enhancement for noises with high brightness, resulting in a high false-alarm rate. To solve this problem, Han et al. proposed an improved Local Contrast Measure (ILCM) method [10]. To improve the detection performance in complex backgrounds, Han et al. proposed the NLCM method [11]. Wei et al. proposed MPCM from the perspective of image patch difference [12]. Deng et al. proposed the weighted local difference method (WLDM) [13], which can better separate the real targets from the interfering object. Han et al. proposed a multi-scale relative local Contrast ratio (RLCM) [14], which can efficiently detect small targets of different sizes in complex backgrounds. Guan et al. proposed the enhanced Local Contrast Measure (ELCM) [15], and Han et al. proposed the weighted enhanced local Contrast (WSLCM) [16] for infrared small target detection. For methods based on background features, Max-Mean and Max-Median [17], based on nonlinear filtering; and classical Top-Hat [18], based on morphological filtering, are traditional. They all use the background difference to obtain small targets. Bai [19] et al. studied new Top-Hat and applied it to infrared small target detection. Zhu [20] et al. combined Top-Hat with low-rank tensor completion to provide a more robust small target detection model. In addition, the two-dimensional least mean square (TDLMS) filter [21] and bilateral filter [22] also have good results using background prediction. However, these methods are not very effective in dealing with complex backgrounds.

With the development of matrix optimization theory, the low-rank sparse matrix recovery method is attracting more and more attention. This method can separate the target image from the background image according to the target’s sparsity and the background’s low rank. The most representative process is the infrared patch-image model (IPI) [23]. Dai et al. proposed a weighted infrared patch-image model (WIPI) [24] to suppress the edges better. The reweighted infrared patch-tensor model (RIPT) [25] and the non-negative infrared patch-image model via partial sum minimization for singular values model (NIPPS) [26] could provide more accurate background estimations. In terms of the matrix recovery improvement, some methods include low-rank sparse representation (LRSR) [27], adaptive weighted parameter [28], total variation regularization, principal component pursuit (TV-PCP) [29], and non-convex rank approximation minimization (NRAM) [30]. Rawat et al. proposed a method via total variation and partial sum minimization (TV-PSMSV) to recognize small targets in a highly complex background [31].

When the background of the infrared image is very complex, such as buildings, mountains, and trees, the low-rank property of the background is very poor, so the conventional low-rank sparse matrix recovery methods are not very effective. In order to resolve the aforementioned problem, this paper proposes a novel infrared small targets detection strategy based on block-matching. The proposed method uses non-local similarity to match and group image blocks to construct a new infrared model. Firstly, the original infrared image is divided into some reference blocks. Similar blocks are searched throughout the image for each reference block, and several block-matching groups are formed. Secondly, each block-matching group is formed into a matrix. We suppose the infrared background image is a low-rank matrix, and that the infrared target image is a sparse matrix. The small target detection problem is transformed into an optimization problem of recovering the low-rank and sparse matrices. Finally, the target image and background image are obtained via reconstruction. A simple threshold segmentation method is used to obtain a small target. Experimental results show that the proposed method has better stability under different backgrounds and better detection performance than the traditional baseline method.

The remainder of this paper is organized as follows. In Section 2, we introduce the process of the new infrared block-matching model construction and reconstruction, and we also analyze the model. Section 3 describes the small target detection method based on block-matching. In Section 4, we present the experimental results of block size effects and comparison experiments in this paper. The conclusion is given in Section 5.

## 2. Infrared Block-Matching Model

In natural images, self-similarity generally exists between image blocks [32]. Many classical image denoising algorithms have been proposed by utilizing the self-similarity of images [33,34,35]. The distance between blocks usually expresses the similarity between two blocks. The smaller the distance between the two blocks, the higher the similarity. Block-matching is a common method in image denoising and motion estimation. It calculates the distance between a reference block and a block to be matched, finds some matching blocks closest to the reference block, and divides them into a set. Figure 1 shows some reference blocks in the image and matched blocks with high similarity.

### 2.1. Construction of Infrared Block-Matching Model

We construct the infrared block-matching model from an original image. The steps are as follows. Firstly, an input infrared image is clipped into several reference blocks. Then, for each reference block, a sliding window is used to search for similar blocks with the reference block from left and top to right and down. Block-matching is a method for finding blocks similar to a given reference one [34]. If the distance between the blocks and the reference one is smaller than a given threshold, they are mutually similar. Therefore, only blocks whose distance concerning the reference one is smaller than a threshold are deemed similar. In particular, we use the $\ell^{2}$-distance as a measure. The block distance can be calculated as: (1)$$d\left(Z_{xr},Z_{x}\right)={\left\|\;Z_{xr}-Z_{x}\;\right\|}_{2}^{2},$$
where $\|\;\cdot {\|}_{2}$ denotes the $\ell^{2}$-norm, and $Z_{xr}$ and $Z_{x}$ respectively represent the reference block and the block to be matched.

We can obtain the set composed of similar blocks from $d(Z_{xr},Z_{x})$: (2)$$S=\{d\left(Z_{xr},Z_{r}\right)<\tau \},$$
where τ is a constant representing the maximum matching threshold of similar blocks. Finally, for each set, we transform each block into a column vector to form a new matrix.

### 2.2. Analysis of Infrared Block-Matching Model

A single infrared image is usually considered to be composed of three parts: (3)$$F_{d}=F_{b}+F_{t}+F_{n},$$
where $F_{d}$, $F_{b}$, $F_{t}$, and $F_{n}$ are the original image, background image, target image, and noise image, respectively. By constructing the infrared block-matching model for the infrared input images, we can transform the infrared image model into the corresponding infrared block-matching image model: (4)D=B+T+N,
where *D*, *B*, *T*, and *N* respectively represent the original image, background image, target image, and noise image in the infrared block-matching model.

Target block-matching T: Small targets are tiny in infrared images. Due to the low proportion of small targets, target block-matching T can be considered a sparse matrix. That is: (5)$$\|\;T{\|}_{0}\leq k,$$
where $\|\;\cdot {\|}_{0}$ denotes the $\ell^{0}$-norm and *k* is a constant determined by the size and number of the small target.

Background block-matching B: As previously discussed, the infrared background image usually has strong non-local self-similarity. We select three representative infrared background images. Additionally, the IBM model matrix is constructed for a reference block of each image. The reference block size is 20 × 20. Meanwhile, we calculate the singular value of the matrix. Figure 2 shows that their singular values all rapidly decrease to zero. Therefore, we can consider background block-matching B as a low-rank matrix. That is: (6)Rank(B)≪r,
where *r* is a constant representing the complexity of the background image. The more complex the background, the greater the value of *r*.

Noise Block-Matching N: In this paper, we assume that the noise is additive white Gaussian noise and $\|\;N{\|}_{F}\leq \delta$ for some δ>0. Thus, we have: (7)$$\|\;D-B-T{\|}_{F}\leq \delta ,$$
where $\|\;\cdot {\|}_{F}$ denotes the Frobenius norm.

Although the parameters *k*, *r*, and δ vary for different infrared images, we would not use them directly in the following sections. We will discuss this further in Section 3.1.

### 2.3. Reconstruction of Infrared Images from the Infrared Block-Matching Model

We reconstruct an image from an infrared block-matching model. The steps are as follows. Firstly, the similar blocks with each reference block are returned to their original positions in the infrared image. Because the selected similar blocks may overlap, a pixel location would correspond to several values from different blocks. In this case, we use a 1D mean filter function to determine the final value of each pixel: (8)V=mean(x),
where V∈R and *x* is a vector containing the values of all similar blocks at one pixel.

## 3. Small Target Detection Based on Block-Matching

### 3.1. Solution of Infrared Small Target Detection

According to Section 2, we can consider the target block-matching T as a sparse matrix and the background block-matching B as a low-rank matrix. Ideally, we transform the small target detection problem based on block-matching into the following low-rank and sparse decomposition problem: (9)$$\min_{B,T}rank\left(B\right)+\lambda \|T{\|}_{0},s.t.B+T=D,$$
where λ denotes the equilibrium factor; we choose the robust principal component analysis (RPCA) to solve (9) because it has better anti-noise interference ability and robustness. Because the parameters k and r in (5) and (6) vary for different infrared images, it is difficult to estimate them in advance. Robust principal component analysis can accurately and efficiently recover B and T in (9) under broad conditions, such as no relevant information about the rank of B or the support of T being given [36]. Therefore, our method does not need to obtain the parameters k and r of different images. Unfortunately, (9) is a nondeterministic polynomial hard (NP-hard) problem [37]. This problem cannot be solved using exact algorithms. Therefore, we need to find an efficient approximation algorithm. Here, $\|\;B{\|}_{*}$ and $\|\;T{\|}_{1}$ replace rank(B) and $\|\;T{\|}_{0}$, respectively. Then, we consider the situation of noise in an image. The above objective function can be transformed into the following convex optimization problem: (10)$$\min_{B,T}\|\;B{\|}_{*}+\lambda \left\|\;T{\|}_{1}s.t.\right\|\;D-B-T{\|}_{F}\leq \delta .$$

The traditional methods to solve the standard RPCA model are iterative thresholding (IT) [36] and accelerated proximal gradient (APG) [38]. In addition, exact augmented Lagrange multipliers (EALM) [39] and inexact augmented Lagrange multipliers (IALM) [40] are also commonly used. We choose IALM to solve (10) because of its faster operation speed and higher operation accuracy.

Construct the augmented Lagrange function for (10): (11)$$\Gamma \left(B,T,Y,\mu \right)=\|\;B{\|}_{*}+\lambda \|\;T{\|}_{1}+\langle Y,D-B-T\rangle +\frac{\mu }{2}{\left\|\;D-B-T\;\right\|}_{F}^{2},$$
where $Y\in R^{m\times n}$ denotes the Lagrange multiplier, μ>0 is the penalty parameter, and 〈•〉 denotes the standard inner product. μ is varied while solving the above optimization, and (10) is equivalent to (11) for some value μ(δ) [41]. When $Y=Y_{K}$ and $\mu =\mu_{k}$, the optimization problem $\min_{B,T}=L\left(B,T,Y,\mu_{k}\right)$ is solved using alternate direction methods.

### 3.2. Target Segmentation

After obtaining the target image, we need to extract a small target from the image. We adopt the same target segmentation threshold as [23]. In this paper, we only detect bright targets. The adaptive threshold $t_{up}$ is defined by: (12)$$t_{up}=\max\left(\upsilon_{\min},\mu +k\sigma \right),$$
where μ and σ respectively denote the average and standard deviation of the target image, and *K* and $V_{min}$ are constants determined experientially.

### 3.3. The Entire Procedure of Small Target Detection

Figure 3 shows the whole process of the small target detection method proposed in this paper. The steps are: (1) According to Section 2.1, we construct the block-matching D from the input infrared image. (2) We use IALM to decompose the block-matching D into the low-rank background block-matching B and sparse block-matching target T. (3) According to Section 2.3, we reconstruct the target image $F_{t}$ and background image $F_{b}$. (4) We obtain the final image by using a simple threshold segmentation on the target image.

## 4. Experiments

### 4.1. Experimental Settings

Evaluation Metrics: In this paper, we choose background suppression factor (BSF), local signal-to-noise ratio gain (LSNRG), and signal-to-clutter ratio gain (SCRG) to evaluate the performance. Three evaluation metrics are calculated in a local region concerning the target, as illustrated in Figure 4. The target size is a×b and the neighborhood width is *d*, which is a constant. We set d=20 in this paper. BSF represents the inhibitory ability to the background. The higher the value of BSF, the better the background suppressing performance. BSF is defined as follows: (13)$$BSF=\frac{\sigma_{in}}{\sigma_{out}},$$
where $\sigma_{in}$ and $\sigma_{out}$ represent the local background neighborhood standard variances of the input and output images, respectively. LSNRG measures the local signal-to-noise ratio (LSNR) gain, where a higher value of LSNRG denotes a better performance. LSNR is defined as follows: (14)$$LSNR=\frac{P_{T}}{P_{B}},$$
where $P_{T}$ and $P_{B}$ represent the maximum target pixel value and the maximum local background neighborhood pixel value, respectively.
(15)$$LSNRG=\frac{LSNR_{out}}{LSNR_{in}},$$
where $LSNR_{out}$ and $LSNR_{in}$ stand for the LSNR values of the output and input images. SCRG represents the validity of target enhancement. The stronger the target enhancement ability, the larger the value. SCR is defined as follows: (16)$$SCR=\frac{\left|\mu_{t}-\mu_{b}\right|}{\sigma_{b}},$$
where $\mu_{t}$ is the average pixel value of the target, and $\mu_{b}$ and $\sigma_{b}$ are the average pixel value and the standard deviation of the local background neighborhood, respectively.
(17)$$SCRG=\frac{SCR_{out}}{SCR_{in}},$$
where $SCR_{out}$ and $SCR_{in}$ stand for the SCR values of the output and input images. In addition to the three evaluation indicators, the receiver operating characteristic (ROC) curve is also an important metric. The ROC curve can directly reflect the accuracy of detection and help to find the appropriate threshold. For small target detection, the larger the area enclosed by the curve and abscissa, the better the method’s performance. Additionally, the higher detection probability for the same false-alarm ratio means a better performance. The ROC curve is plotted using the detection probability $P_{d}$ and the false-alarm rate $F_{a}$. $P_{d}$ and $F_{a}$ are defined as follows: (18)$$P_{d}=\frac{\mathrm{number}\;\mathrm{of}\;\mathrm{true}\;\mathrm{detections}}{\mathrm{number}\;\mathrm{of}\;\mathrm{actual}\;\mathrm{targets}}$$
(19)$$F_{a}=\frac{\mathrm{number}\;\mathrm{of}\;\mathrm{false}\;\mathrm{pixels}\;\mathrm{detected}}{\mathrm{number}\;\mathrm{of}\;\mathrm{pixels}\;\mathrm{in}\;\mathrm{all}\;\mathrm{images}}.$$

Baseline Methods: In this paper, we choose Max-Mean [17], Max-Median [17], Top-Hat [18], IPI [23], and TV-PCP [29] as the baseline methods. The detailed parameters of these methods are shown in Table 1.

Experimental Datasets: In this paper, we select four infrared sequences for the experiments. All sequences are from public datasets [42]. Detailed information on the infrared sequence is shown in Table 2.

### 4.2. Effect of Block Size

In our method, the size of the reference block is an important parameter. Different block sizes will affect the algorithm performances for different datasets. Thus, we need to adjust it to obtain better detection results. We set the square block size as 5 × 5, 10 × 10, 20 × 20, and 50 × 50. Figure 5 shows the separated target images according to the different sizes of the reference block. We adjust the contrast of the images for clearer viewing. As seen from Figure 5, the smaller the block size, the better the background suppression performance, and the less the background residue of the target images for the four sequences. This is because the smaller the block size, the higher the matching degree between the matching block and the reference block. This results in a sparser image of the block-matching model when using a smaller block size. However, as the image of the constructed block-matching model is sparser, the low-rank property of targets will be lost to a certain extent. Therefore, the contrast between the target and background also decreases.

To further discuss the effect of block size, we display the ROC curve for different block sizes. The ROC curves for the four sequences are shown in Figure 6. It can be seen from Figure 6 that the ROC curves of different sequences under different block sizes have little differences. In particular, for Seq.2 and Seq.3, the smaller the block size, the better the performance. However, the pattern is not evident for Seq.1 and Seq.4. This is because Seq.2 and Seq.3 have a larger target size than Seq.1 and Seq.4. According to the above analysis, the low-rank property of the target will be lost to a certain extent. As the size of the target becomes bigger, this effect becomes more pronounced. It should be noted that the smaller the block size, the longer it takes to obtain the separated target images. In combination with the above qualitative and quantitative analysis, we choose block size 10 × 10 as the parameters of our method in the subsequent contrast experiments, based on the consideration of computational cost.

### 4.3. Contrast Experiments

In order to verify the effectiveness of the proposed method, baseline methods including Top-Hat, Max-Mean, Max-Median, IPI, and TV-PCP are selected for comparison. We select a representative image from each of the four sequences and process them with the six methods. Firstly, we compare the background suppression performance. The detection results of the representative images of four sequences using different methods are shown in Figure 7.

It can be seen in Figure 7 that Top-Hat performs very poorly in the face of four sequences with complex backgrounds. It can enhance the target. However, because Top-Hat relies on the assumption of the size of target, it cannot distinguish the target from the background clutter well. At the same time, it can not screen noise that is close to the target size. Max-Mean and Max-Median can enhance the target, but the ability to suppress background clutter is weak. Max-Median is more robust than Max-Mean. According to the results of Seq.2 and Seq.4, it can be seen that Max-Median has a better inhibition effect on strong edges compared to Max-Mean. The common reason for why Top-Hat, Max-Mean, and Max-Median do not perform well is that they need to make preset assumptions about the target information. The last IPI, TV-PCP, and our method are all based on low-rank sparse matrix recovery. IPI enhances the background sparsity by constructing the patch-image. Therefore, the result is good in the face of the sequence (Seq.2), with a relatively smooth background transition. When the background is very complex with a lot of highlighted interference, the correlation between adjacent pixels in the image background is not strong. This results in a worse IPI performance in Seq.1, Seq.3, and Seq.4. TV-PCP brings TV regularization term to the traditional PCP model. It can better deal with non-uniform images. However, TV-PCP performs poorly when faced with strong edges and highlighted backgrounds such as IPI. This is because they are all based on the low-rank property of the constructed patch-image. In contrast, the proposed method in this paper is better at suppressing the nontarget components. Our method has the least residual background clutter in the experimental images, as it makes full use of the self-similarity of whole image. Although there is a very small amount of clutter because of the complex background, we can further use threshold segmentation to obtain small targets. Based on the above comparisons, the proposed method achieves the best background suppression effect among the six tested methods.

We calculate the metrics values without threshold segmentation to further evaluate these methods, and higher values denote better performance. The performance comparison of the six methods is shown in Table 3. Table 3 shows that the proposed method yields the best or second-best results for most of the image sequences, compared with the other methods. This suggests that the proposed method in this paper outperforms the baseline methods in terms of background suppression and target enhancement.

Finally, to further reveal the advantage of the proposed method, we display the ROC curve to reflect the methods’ performance more intuitively. The ROC curves are shown in Figure 8. As seen in Figure 8, the detection rate of the proposed method is more prominent. On the whole, it has a higher detection probability under the same false-alarm rate. In Seq.2, IPI, TV-PCP, and our method have achieved good results due to the relatively smooth background. TV-PCP performs well in Seq.1. However, the false-alarm rate was higher in Seq.3 and Seq.4, with more highlighted points. IPI also performs well in Seq.3 and Seq.4, but the detection rate is not as good as our method due to the poor ability of the algorithm to deal with the non-uniform background. Top-Hat, Max-Median, and Max-Mean all had unsatisfactory results due to their dependence on the assumption of target information. In particular, Max-Median does badly in Seq.4. Our method shows better detection performance when dealing with complex backgrounds.

## 5. Conclusions

This paper proposed an infrared small target detection method based on block-matching in images. This method takes full advantage of non-local similarity to construct an infrared block-matching model. Then, the small target detection task is transformed into an optimization problem of recovering low-rank and sparse matrices. This problem can be effectively solved via IALM. Our method reduces the dependence on the correlation between adjacent pixels in the image background. When the background of images has strong edges and highlighted clutter, the block-matching model has better low-rank property. This block-matching model can significantly suppress the background clutter and noise to improve the performance of small target detection. The experimental results show that our proposed method not only obtains the clearest separated target images compared with the baseline methods, but also significantly improves quantitative parameters such as LSNRG, SCRG, BSF, and the ROC curve. In the future, we will study the feasibility of the block-matching model in 3D or multi-dimensional space. We will also learn the matrix recovery algorithm applicable to the proposed method further to improve the speed and applicability.

## Figures and Tables

**Figure 1 sensors-22-08300-f001:**
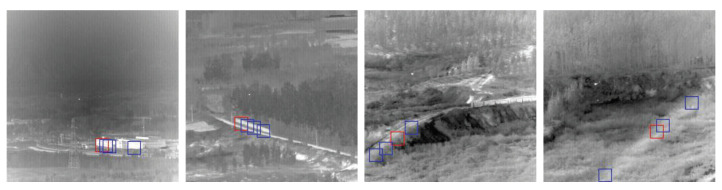
Illustration of matching blocks of infrared images. Each image shows a reference block marked with red borders and a few of the blocks marked with blue borders matched to it.

**Figure 2 sensors-22-08300-f002:**
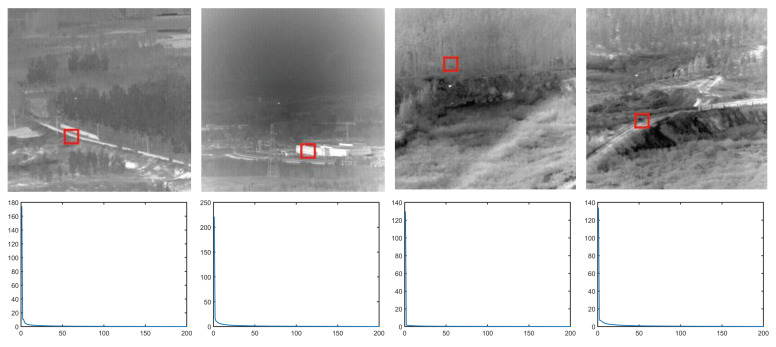
Illustration of the low-rank property of background block-matching B. The first row are four images, each with a background reference block with a red border, and the second one are the singular values of background block-matching B constructed using four reference blocks.

**Figure 3 sensors-22-08300-f003:**
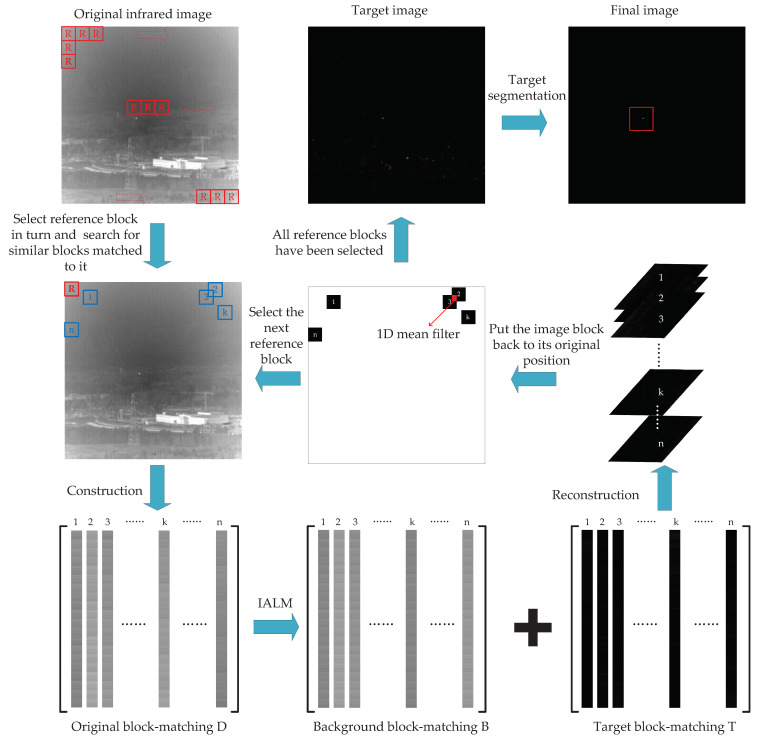
The overview of the proposed method in this paper.

**Figure 4 sensors-22-08300-f004:**
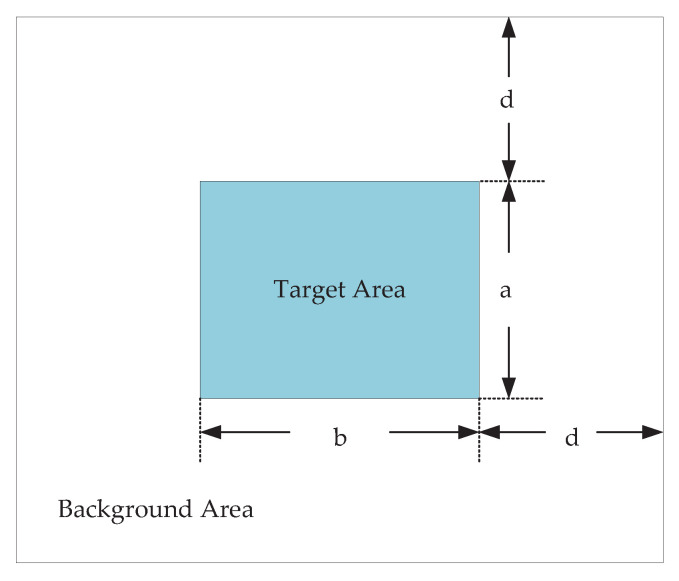
Target area and background neighborhood area of the small target.

**Figure 5 sensors-22-08300-f005:**
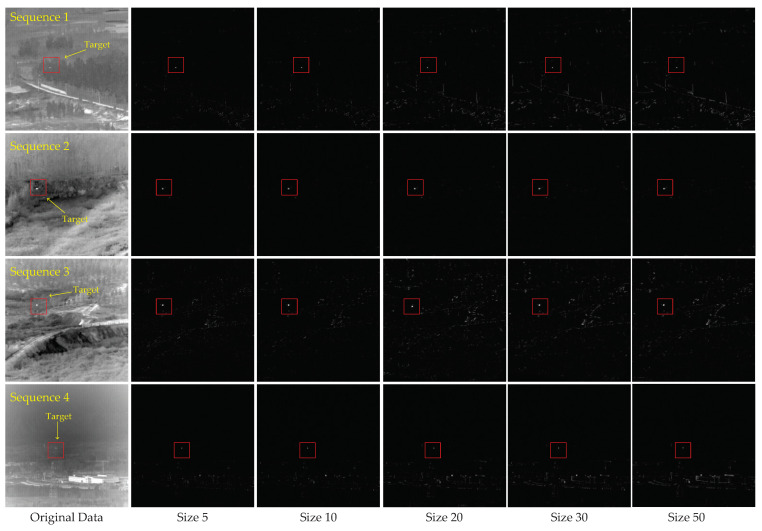
The representative images of the four real image sequences and the corresponding processed target images of different block sizes. From left to right are the original data and the processing results of size 5, size 10, size 20, size 30, and size 50.

**Figure 6 sensors-22-08300-f006:**
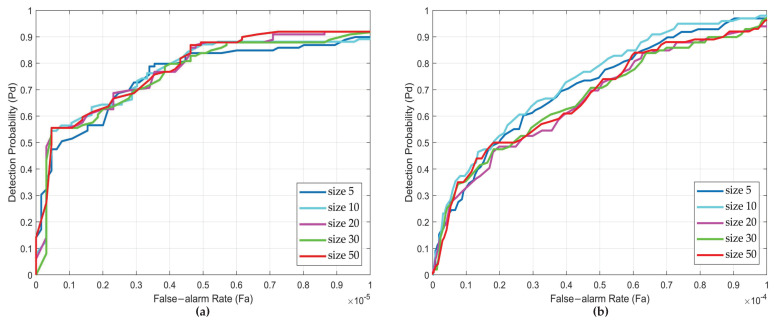
The receiver operating characteristic (ROC) curves of different sizes for four real image sequences. (**a**) Sequence 1. (**b**) Sequence 2. (**c**) Sequence 3. (**d**) Sequence 4.

**Figure 7 sensors-22-08300-f007:**
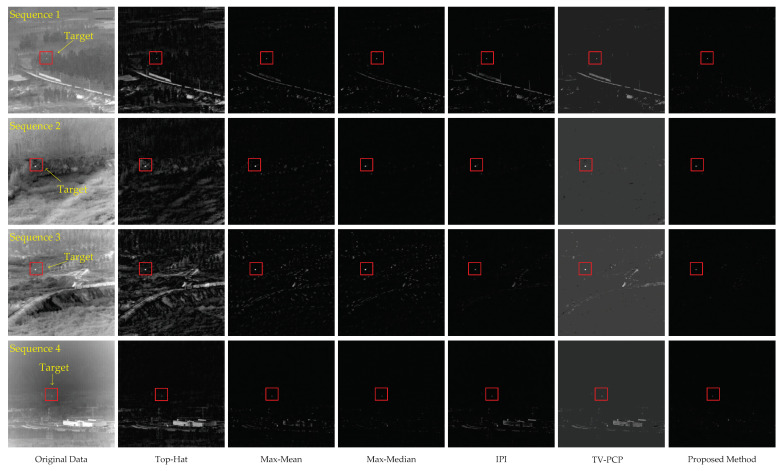
The representative images of the four real image sequences and the corresponding processed target images of different methods. From left to right are the original data, the processing results of Top-Hat, Max-Mean, Max-Median, IPI, TV-PCP, and the proposed method.

**Figure 8 sensors-22-08300-f008:**
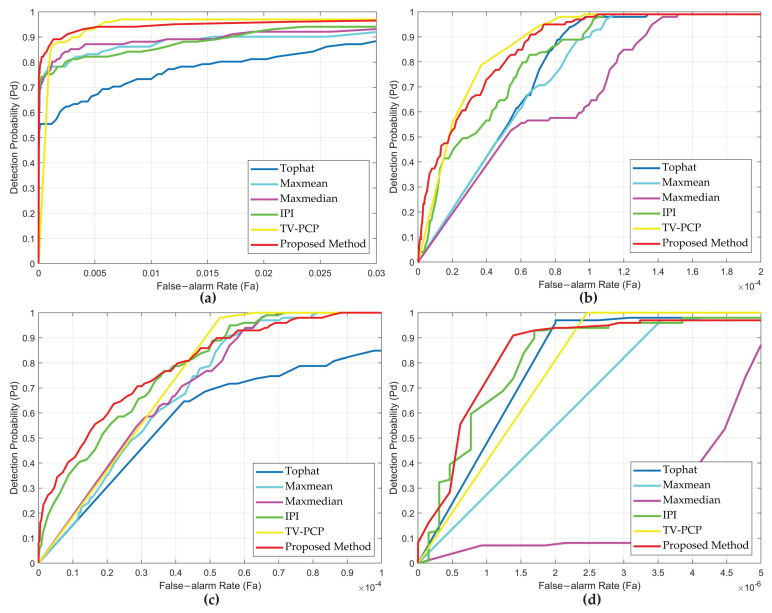
The receiver operating characteristic (ROC) curves of different methods for four real image sequences. (**a**) Sequence 1. (**b**) Sequence 2. (**c**) Sequence 3. (**d**) Sequence 4.

**Table 1 sensors-22-08300-t001:** Detailed parameters of baseline methods.

Methods	Parameters
Max-Mean [17]	Filter size: 15 × 15
Max-Median [17]	Filter size: 15 × 15
Top-Hat [18]	Shape: square, filter size:15 × 15
IPI [23]	Patch size: 50 × 50, sliding step: 10, λ = $\frac{1}{\sqrt{\min(m,n)}}$, $\varepsilon =10^{-7}$
TV-PCP [29]	Patch size: 50 × 50, sliding step: 14, λ=0.005, $\lambda_{2}$ = $\frac{1}{\sqrt{\min(m,n)}}$, β=0.025, γ=1.5, maxIter = 250, $tol=5\times 10^{-6}$

**Table 2 sensors-22-08300-t002:** Detailed description of experimental sequences.

N0.	Frame	Image Size	Characteristics
1	200	256 × 256	Complex background with trees and highlighted interference
2	200	256 × 256	Complex background with thickets
3	200	256 × 256	Complex ground background with trees and road
4	200	256 × 256	Complex background with constructions

**Table 3 sensors-22-08300-t003:** Quantitative comparison of six methods for the images of Sequence 1–4.

	Top-Hat	Max-Mean	Max-Median	IPI	TV-PCP	Proposed Method
10th frame of Seq.1						
BSF	1.9150	2.2114	2.2241	2.7063	2.8176	**2.9542**
LSNRG	1.8555	2.4063	2.2001	2.8511	2.3334	**3.3821**
SCRG	1.8726	3.6916	4.1848	4.5444	4.6315	**4.8301**
10th frame of Seq.2						
BSF	1.4953	1.7571	2.1528	1.9926	1.8748	**2.2620**
LSNRG	1.0046	0.8935	**1.0860**	1.0236	1.0116	1.0704
SCRG	1.3216	2.3538	2.8490	2.6366	**3.1674**	2.8290
10th frame of Seq.3						
BSF	1.4840	1.5812	1.5812	1.8878	1.3768	**1.9611**
LSNRG	1.1988	1.3980	1.1080	1.4019	1.2662	**1.4470**
SCRG	1.0564	2.3859	2.4129	3.1280	2.8973	**3.1363**
10th frame of Seq.4						
BSF	1.7803	2.6111	2.8659	**3.0519**	2.3717	2.9747
LSNRG	2.0182	**3.0833**	2.4292	2.5882	1.4800	2.5807
SCRG	1.6963	3.9429	4.0535	**4.3271**	4.1347	4.2013

## Data Availability

Not applicable.

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
