# Peer review of "Infrared Single-Frame Small Target Detection Based on Block-Matching"

_sensors, 2022, doi:10.3390/s22218300_

Round 1
Reviewer 1 Report
Markings:
1. Infrared single frame detection method based on the block-matching approach is proposed in this paper
2. A new infrared model is constructed by finding blocks that are similar to each such block. Then, the small target detection based on the block-matching model is formulated as an optimization problem of recovering low-rank and sparse matrices, which are effectively, solved using robust principal component analysis.
3. A simple segmentation method is used to segment the target image. The experimental results from actual infrared sequences show that the proposed method has better background suppression ability under complex backgrounds and better detection performance than conventional baseline methods.
4. I feel to consider this manuscript for publication in this esteemed journal as in its current form.
Reviewer 2 Report
A summary:
This manuscript demonstrates an infrared single-frame detection method based on the blocking-matching approach. The topic is interesting. However, this manuscript sounds more like a report than a scientific article. The figures, tables, and formulas are simply listed and are not explained further, which makes it difficult for the readers to understand. Some of the expressions are either wordy or too concise. The figures are not of good quality. The conclusion part is too short and the limitations of the methods in this paper are not mentioned. The following are my detailed comments and I hope that those will be helpful in improving the quality of this manuscript.
Specific comments:
L3: “a infrared system” -> an infrared system.
L29: “chal-lenging” -> challenging
L102: Please provide more explanations for this method.
L111: What is “IBM model”?
L131-132: Can you give further explanations for this sentence?
L148: Please explain what an NP-hard problem is.
L151: “opti-mization” -> optimazation
L160: Since Figure 5 includes the information from Figures 2 and 4, Figures 2 and 4 are unnecessary. What is the difference between the target image and the final image?
L207: Figure 7 needs some work. The subplots with dark backgrounds are difficult to recognize and distinguish from each other.
L248: Do you mean Table 2 or Table 3?
L150-153: These sentences are wordy. Can you rewrite them?
L261: The conclusion part is too short. It does not include the limitations of the research and future work. I recommend that the authors revise it to make it more comprehensive.
L262: “a infrared” -> an infrared
Reviewer 3 Report
This paper developed an infrared single frame detection method based on the block-matching approach. The small target detection task is then transformed into an optimization problem that can be effectively solved. The theoretical derivation of this method is Sec. 2 and 3 is rigorous. Through the experimental tests, this proposed method successfully obtains the explicit separated target images in the face of the complex highlighted background. The comparisons of several baseline methods demonstrates that the proposed method achieves a better performance. The experimental analysis and the corresponding results in Sec. 4 are correct. It is a good job and I suggest this paper to be accepted in the present form.

Reviewer 4 Report
The manuscript describes a method for infrared signle-frame small target detection. The topic of the manuscript is timely and interesting, with the importance to the readers interested in the specific research field. The introduction provides a solid background and includes relevant references. The description of the small target detection based on block-matching is concise, and the presentation of the experimental settings and results is clear. In my opinion, the manuscript can be published after minor revisions:
- spelling and grammar should be checked and improved/corrected where needed, for example:
- line 26: space is missing after the bracket,
- line 29: correct "chal-lenging",
- lines 47 and 49: space is missing after the punctuation mark,
- lines 83 and 86: add space after "Section",
- line 140: "Block" instead of "Blcok",
- line 176: add space after ":".
Why did the authors choose specific baseline methods listed in lines 196 and 197?
Conclusion should be reflecting the specificity of the results and scientific contribution that this manuscript provides - it is a bit too vague.
Reviewer 5 Report
In the present manuscript the authors report infrared single-frame small target detection. They have adopted block matching method. After going though it I found that the paper lacks clarity and conclusiveness. Results are not explained to decipher the novelty of the approach. Hence I cannot recommend it for publication in its present form.
Points of concern are
The results shown in the figures are not explained in details and with clarity. It is not clear what they want to convey from the figures.
Figures need far better clarity to understand the effects.
Figure captions should be detailed and self-explanatory. Unfortunately these are just terse statements.
Fig. 10, why different methods have different behaviours in different sequences?
L219, 220; they mention “For Seq.1 and Seq.3, the smaller the block size, the better the performance. However, the pattern is not evident for Seq.1 and Seq.4”. There is no explanation for it.
L102, “ROC curve is plotted by detection probability Pd and1false-alarm rate Fa. Its significance needs to be pointed out.
Critical Comparison with a recent paper on the same theme “Mathematics 2022, 10, 671. https://doi.org/10.3390/math10040671” has to be there.
In the conclusion they mention that “Extensive experimental results show that our method is more robust and efficient” A critical comparison with other methods is explicitly needed to arrive at this statement.
Minor language related problems have also to be taken care; particularly with the use of article ’the’.
Round 2
Reviewer 2 Report
I have read the revised manuscript and found the authors did a good job in addressing most concerns pointed out by the reviewers in the first round of reviews. The quality of this manuscript has been improved. I recommend publication of this manuscript in its present form.
Author Response
Thank you for your valuable and thoughtful comments. Regarding the questions about English writing, We have had our manuscript checked by a colleague fluent in English writing, and we have carefully reviewed and improved the English writing in the revised manuscript. Thank.
Reviewer 5 Report
In this revised version, the authors have addressed the technical comments satisfactorily. However, I still notice some language related issues which have to be fixed before publication. Some are as follows-
L 207, 'the' should come before value and background; L 218 'is' to be removed after ability; L224, 'the' should come before area; L 244 , 'T in The" should be in small letter; L246,word 'found' is not needed; L249, 264, be is better word in place of 'become'; L261, Seq.1 appears incorrect; L289, 'different' is not proper word; L332, i should be in small letters in 'It';
